# Transcriptomic Analysis Reveals the Flavonoid Biosynthesis Pathway Involved in Rhizome Development in *Polygonatum cyrtonema* Hua

**DOI:** 10.3390/plants13111524

**Published:** 2024-05-31

**Authors:** Kui Wan, Jingjie Ban, Fengjie Yang, Xueying Zhang, Xiaoling Huang, Yanqiu Wang, Zihao Zhang, Zhongxiong Lai, Yukun Chen, Yuling Lin

**Affiliations:** Institute of Horticultural Biotechnology, Fujian Agriculture and Forestry University, Fuzhou 350002, China; 15565393028@163.com (K.W.); 42065857@126.com (J.B.); zxyyingg@163.com (X.Z.); hxl12072022@163.com (X.H.); 18259963009@163.com (Y.W.); zhangzihao863@126.com (Z.Z.); laizx01@163.com (Z.L.)

**Keywords:** *Polygonatum cyrtonema* Hua, rhizome development, flavonoid biosynthesis, transcriptomics, Illumina HiSeq

## Abstract

*Polygonatum cyrtonema* Hua (*P. cyrtonema*) rhizomes are rich in flavonoids and other secondary metabolites, exhibiting remarkable antioxidant, anti-tumor, and immunomodulatory effects. *Polygonatum* flavonoid-biosynthesis-related genes have been characterized already. However, a comprehensive overview of *Polygonatum* flavonoid biosynthesis pathways is still absent. To articulate the accumulation of the flavonoid biosynthesis pathways, we examined transcriptome changes using Illumina HiSeq from five different tissues and the RNA-seq of 15 samples had over 105 Gb of a clean base, generating a total of 277,955 unigenes. The cDNA libraries of the fruits (F), leaves (L), roots (R), stems (S), and rhizomes (T) of three-year-old *P. cyrtonema* plants generated 57,591, 53,578, 60,321, 51,530, and 54,935 unigenes. Comparative transcriptome analysis revealed that 379 differentially expressed genes (DEGs) were in the group of F _vs_ T, L _vs_ T, R _vs_ T, and S _vs_ T, and the transcripts of flavonoid-biosynthesis-related DEGs were principally enriched in rhizomes. In addition, combined with WGCNA and the FPKM of five tissues’ transcription, nine differentially expressed transcription factor families (*MYB*, *WRKY*, *AP2/ERF*, etc.) were characterized in the red module, the red module positively correlated with rhizome flavonoid accumulation. Quantitative real-time PCR (qRT-PCR) further indicated that *BZIP1*, *C3H31*, *ERF114*, and *DREB21* are differentially expressed in rhizomes, accompanied in rhizome development in *P. cyrtonema*. Therefore, this study provides a foundation for further research into uncovering the accumulation of flavonoid biosynthesis in the rhizomes of *P. cyrtonema*.

## 1. Introduction

*Polygonatum cyrtonema* Hua (*P. cyrtonema*) is an important Chinese herb, possessing a homology between medicine and food with respect to *Polygonatum Mill* (Liliaceae), which is included in the Chinese Pharmacopoeia as a basal species of the Chinese medicine *Polygonatum*, together with *P. sibiricum* and *P. kingianum*. There are various *Polygonatum Mill* species abundantly and widely distributed in the north temperate zone, and there are 31 species of *Polygonatum* in China [1,2]. The rhizome is a vital storage organ in *P. cyrtonema*, with high metabolic process. Flavonoids and polysaccharides are the main medicinal ingredients of *P. cyrtonema* rhizomes [3]. and also serve as important indicators of the herb quality of *P. cyrtonema* rhizomes [4].

Recently, with the development of genomics, transcriptome-sequencing technology has been widely used. Multiple plants have been analyzed via the use of transcriptome-sequencing technology to study gene functions, screen molecular markers [5], and unravel metabolic pathways; these plants include *Fagopyrum tataricu* [6], *Salvia miltiorrhiza Bunge* [7], *Panax ginseng* [8], and other species. To date, most of the transcriptome-level studies of flavonoids have focused on *P. kingianum* and *P. sibiricum*, mostly in terms of the biosynthesis pathways of polysaccharides and steroidal saponins [9,10]. Due to the lack of genomic information, the mechanism of molecular regulation involved in *P. cyrtonema* flavonoid biosynthesis pathways is still unclear. Ye [11] used three- and nine-month rhizomes of *P. cyrtonema* to perform transcriptomic analysis. Consequently, 12 key genes involved in flavonoid biosynthesis pathways were identified. The number and type of flavonoid-biosynthesis-related genes identified differed from those identified in other studies [12,13,14], which might have been because the expression of flavonoid-biosynthesis-related genes differed in the leaves, flowers, and stems during rhizome development. Chai [15] analyzed the transcriptomics of *P. kingianum* leaves and found 368 genes involved in flavonoid biosynthesis, among which were 6 differentially expressed genes. Xiao [16] used six-month-old roots, stems, and leaves of *P. kingianum* for transcriptomic analysis, revealing ten differential flavonoid-biosynthesis-related genes that were screened in *P. kingianum*, and the expression of *CHI* and *CHS* determined the flavonoid content.

Flavonoids are secondary metabolites displaying a “flavan” skeleton. They are widely distributed in plants and have pharmcological effects such as hypoglycaemic, anti-inflammatory, antioxidant, anti-tumor, and anti-aging properties [17,18]. To date, hyperisoflavonoids, isoflavones, flavonoids, chalcones, dihydroflavonoids, and zitanes have been isolated from eight species of *Polygonatum*, and hyperisoflavonoids are characteristic components of *Polygonatum* [19]. Furthermore, flavonoid biosynthesis pathways have been reported in *Arabidopsis thaliana* and *rice* [20,21,22,23]. Therefore, it is crucial to study the regulation mechanism of the flavonoid biosynthesis pathway of *Polygonatum*.

Transcription factors (TFs) are key agents that regulate the expression of structural genes at the transcriptional level. Studies have shown that *MYB*, *bHLH*, *WD40*, *NAC*, and *AP2/ERF* are involved in flavonoid biosynthesis [24,25,26,27,28,29,30,31,32,33,34]. The MYB family of transcription factors is one of the largest families of TFs. It is widely present in plants and involved in almost all aspects of plant development and metabolism. *FaMYB5* is involved in the formation of the *MYB–bHLH–WD40* complex (MBW) and positively regulates flavonoid biosynthesis [35]. The promotion of anthocyanin accumulation by ethylene was attributed to the transcriptional activation of *F3H* and *MYBA* genes by *ERF5* [36].

At present, the lack of genomic information hinders the systematic study of the flavonoid-biosynthesis-related genes of *P. cyrtonema*. To elucidate the accumulation of flavonoid biosynthesis pathways in the rhizomes of *P. cyrtonema*, we conducted a transcriptomic analysis of different tissues, including, fruits, leaves, roots, stems, and rhizomes, taken from three-year-old *P. cyrtonema* plants using Illumina RNA-seq technology and the flavonoid content in different age section rhizomes and different tissues, indicated the accumulation of flavonoid may be involved in rhizome development in *P. cyrtonema*. Meanwhile, combined with WGCNA and the FPKM of five tissues’ transcription, quantitative real-time PCR (qRT-PCR) further indicated that *BZIP1*, *C3H31*, *ERF114*, *DREB21*, and *DREB12* may be involved in rhizome development in *P. cyrtonema*.

## 2. Results

### 2.1. Statistics on the Quality of Transcriptome Sequencing and Assembly Results for P. cyrtonema

Fifteen cDNA libraries were constructed for the fruit (F-1, F-2, and F-3), leaves (L-1, L-2, and L-3), roots (R-1, R-2, and R-3), stems (S-1, S-2, and S-3), and rhizomes (T-1, T-2, and T-3) of *P. cyrtonema* (Figure 1A–F). Each library contained three biological replicates, and the cDNA libraries were sequenced using the IlluminaHiSeq high-throughput sequencing platform. The number of raw reads generated for each cDNA library varied between 43,466,000 and 60,504,946. The percentage of Q20 (corresponding to a sequencing error rate less than 1%) was greater than 97%, and GC content was within 45–49% (Appendix A). Pearson’s correlation analysis conducted between the different transcriptome samples and principal component analysis performed on the different samples indicated significant differences between the different tissues of *P. cyrtonema* (Appendix A). All clean data were reassembled using the Trinity program, resulting in a total of 300,667 transcripts and 277,955 genes, with average lengths of 721 bp and 760 bp and N50 lengths of 1107 bp and 1164 bp, respectively. In total, 20.7% of the transcripts and 22.4% of the genes exhibited lengths of >1 kb (Appendix A).

### 2.2. Functional Annotation and Classification of Genes

The constructed 277,955 genes were annotated to seven databases using BLAST; of these genes, 116,511 (41.92%) were annotated to at least one public database, while 79,598 (28.64%), 113,038 (40.67%), 72,391 (26.04%), 108,522 (39.04%), 63,974 (23.02%), 84,345 (30.34%), and 62,697 (22.56%) genes were matched on the KEGG, NR, Swiss-Prot, Trembl, KOG, GO, and Pfam databases, respectively (Figure 2A and Appendix A). The NR database had the largest number of annotations among all the databases, with 57.68%, 6%, and 5.81% of the 113,038 genes in *Asparagus officinalis*, *Elaeis guineensis Jacq*, and *Phoenix dactylifera*, respectively, being annotated (Figure 2B).

A total of 84,345 (30.34%) genes were assigned to one or more GO terms, forming 59 functional groups, which were classified into three categories based on their functions: cellular components, molecular functions, and biological processes. For “cellular component”, the most obvious matches were “cell”, “cell part”, and “organelle”; “cellular processes” and “metabolic processes” were the richest categories of biological processes, while in molecular function terminology, “binding” and “catalytic activity” were rich (Figure 2C). The KEGG database classified the annotated 79,598 (28.64%) genes into 144 pathways. In addition, KOG classified 63,974 (23.02%) genes into 25 categories, with the most common groups being “post-translational modifications, protein turnover, chaperones”, “signal transduction mechanisms”, and “translation, ribosome structure and biogenesis” (Figure 2D).

### 2.3. Differential Expression Gene Analysis Reveals the Flavonoid Biosynthesis Pathways Involved in Rhizome Development in P. cyrtonema

Unigenes with FPKM > 1 in each tissue were counted. The results showed that 28,284, 30,788, 27,583, 29,917, and 27,849 unigenes were expressed in the fruit (F), leaves (L), roots (R), stems (S), and rhizomes (T), respectively. Here, *p*-adjust < 0.05 and |log2^FoldChange^| ≥ 1 were utilized as criteria. In addition, GO, KOG, and KEGG enrichment analyses were conducted on four groups (F vs. T, L vs. T, R vs. T, and S vs. T) of DEGs (differentially expressed genes).

There were 6838 DEGs in F vs. T; of these, 3219 DEGs were upregulated, and 3619 DEGs were downregulated (Appendix A). For the F vs. T group, GO enrichment was mainly seen in relation to “photosystem”, “chloroplast vesicle membrane protein complex”, and “photosynthesis” (Figure 3A1). KOG enrichment was predominantly in relation to “cell cycle control, cell division, chromosome partitioning”; “cell wall/membrane/envelope biogenesis”; and “cell motility” (Figure 3B1). KEGG enrichment was mostly in terms of “carotenoid biosynthesis”, “biosynthesis of secondary metabolites”, “metabolic pathways”, and “flavonoid biosynthesis” (Figure 3C1).

There were 15,507 DEGs in L vs. T; among them, 6479 DEGs were upregulated, and 9028 DEGs were downregulated (Appendix A). For the L vs. T group, GO enrichment was mainly in “inphotosystem”, “thylakoid lumen”, “beta-fructofuranosidase activity”, “sucrose alpha-glucosidase activity”, “flavonoid biosynthetic process”, “terpenoid biosynthetic process”, and “flavonoid metabolic process” (Figure 3A2). KOG enrichment was different in relation to “Secondary metabolites biosynthesis, transport and catabolism”; “cell cycle control, cell division, chromosome partitioning”; and “Signal transduction mechanisms” (Figure 3B2). KEGG enrichment was primarily observed in relation to “isoflavonoid biosynthesis”, “biosynthesis of secondary metabolites”, “carotenoid biosynthesis”, and “flavone and flavonol biosynthesis” (Figure 3C2).

There were 3606 DEGs in R vs. T; 1313 DEGs were upregulated, and 2293 DEGs were downregulated (Appendix A). For the R vs. T group, GO enrichment was mainly in relation to “anchored component of plasma membrane”, “box H/ACA snoRNP complex”, “extracellular region part”, “enzyme inhibitor activity”, “peroxidase activity”, and “fructosyltransferase activity” (Figure 3A3). KOG enrichment was different in terms of “phenylpropanoid biosynthesis”, “biosynthesis of secondary metabolites”, “metabolic pathways”, “flavonoid biosynthesis”, and “anthocyanin biosynthesis” (Figure 3B3). KEGG enrichment was mostly in relation to “cell wall polysaccharide metabolic process”, “phenylpropanoid biosynthetic process”, “lignin biosynthetic process”, “flavonoid biosynthetic process”, and “flavonoid metabolic process” (Figure 3C3).

There were 6847 DEGs in S vs. T; 3319 DEGs were upregulated, and 3528 DEGs were downregulated (Appendix A). For the S vs. T group, GO enrichment was mainly in relation to the anchored “component of plasma membrane”, “senescence-associated vacuole”, “chloroplast thylakoid membrane protein complex”, “fructosyltransferase activity”, “terpenoid biosynthetic process”, and “flavonoid metabolic process” (Figure 3A4). KOG enrichment was different in terms of “cell cycle control, cell division, chromosome partitioning”; “cell wall/membrane/envelope biogenesis”; and “secondary metabolites biosynthesis, transport and catabolism” (Figure 3B4). KEGG enrichment was mostly in relation to “biosynthesis of secondary metabolites”, “phenylpropanoid biosynthesis”, “carotenoid biosynthesis”, and “flavonoid biosynthesis” (Figure 3C4).

In total, for the four groups, GO enrichment mostly corresponded to “fructosyltransferase activity”, “photosynthesis”, and “enzyme inhibitor activity”. KOG enrichment for the four groups was mainly in relation to R (for general function prediction only), O (for post-translational modification, protein flipping, and chaperones) and T (for signal transduction mechanisms). The four groups’ KEGG enrichment was different in relation to “metabolic pathways”, “flavonoid biosynthesis”, “anthocyanin biosynthesis”, and “isoflavonoid biosynthesis”.

Meanwhile, K-means clustering analysis showed that the clustering of genes with high expression was different in the five samples (Figure 3D,E). A total of 22,263 DEGs were identified in the F vs. T, L vs. T, R vs. T, and S vs. T groups; among them, 379 DEGs were in the four groups, and 181 DEGs were highly expressed in rhizomes (Figure 3F), mainly corresponding to “metabolic pathways” and “flavonoid biosynthesis” in the major databases. Additionally, it was speculated that these DEGs may be the key factors for revealing the flavonoid biosynthesis pathways involved in rhizome development in *P. cyrtonema*.

### 2.4. Variation in Flavonoid Accumulation in Different Tissues of P. cyrtonema

Based on the broadly targeted metabolomes of different age section rhizomes (AT: the one-year age section rhizome, BT: the two-year age section rhizome, and CT: the three-year age section rhizome) obtained in the previous stage [37], 27 flavonoids were screened from the metabolome using the secondary classification of the samples (Figure 4A,B). Among the groups, AT, BT, and CT are rich in apigenin and catechin; however, the content and types of flavonoids varied in different age sections. Flavonoid carbon glycosides and flavonols, such as apigenin-6-C-arabinoside, gold sage xanthin-6,8-di-C-glucoside, and tamarix xanthin-3-O-glucoside-7-O-rhamnoside, were rich in AT; dihydroflavonoids and other substances, including coccolithin, azaleatin, and 8-desmethyl azaleatin, were remarkable in BT; glycosidic flavanols and flavonoid carbon sugars, such as gallocatechin gallate, epigallocatechin gallate, apigenin-6-C-xyloside-8-C-arabinoside, apigenin-6-C-arabinoside-8-C-xyloside, apigenin-6, 8-di-C-arabinoside, were outstanding in CT. The flavonoids in the rhizomes of *P. cyrtonema* changed from flavonoid carbon glycosides and flavonols (AT) to dihydroflavonoids (BT) as well as flavanols and flavonoid carbon sugars (CT) with the development of rhizomes.

Meanwhile, the flavonoid content was increasing gradually from the one-year age section rhizome to the three-year age section rhizome (Appendix A). the flavonoid content in different tissues was rich significantly in rhizome (Appendix A). The changes provided data supporting the flavonoid biosynthesis pathways may be involved in rhizome development in *P. cyrtonema*.

Based on transcriptome of five different tissues, metabolome of different age section rhizomes, a total of 149 flavonoid structural genes were screened, revealing 114 DEGs encoding 10 key enzymes, 6 differential flavonoids. Meanwhile, outlining variation in flavonoid accumulation in the different tissues of *P. cyrtonema* (Figure 4D). Quantitative real-time PCR (qRT-PCR) further showed the expression levels of the genes of *4CL3*, *C4H1*, *CHS*, *CHS1*, *CHS2*, and *CHI2*, which were gradually upregulated, while *PAL3*, *C4H2*, *FLS1*, and *FLS2* were downregulated in BT and upregulated in CT, suggesting that *CHI*, *4CL*, and *C4H* play important roles in the flavonoid biosynthesis pathways of *P.cyrtonema* (Figure 4C), the expression of these genes also affected the flavonoid content of rhizome in *P. cyrtonema*.

### 2.5. Affected Rhizome Development Revealed through WGCNA-Based Screening for Flavonoid-Biosynthesis-Related Transcription Factors in P. cyrtonema

The top 100 transcription factor genes (TF) were screened by comparing the PlantTFDB databases, with |log2^FoldChange^| ≥ 1 and FDR < 0.05, in terms of the rhizomes and classified into 34 TF families; of these, 9 TF families were remarkable in the rhizomes, including *MYB*-related, *WRKY*, *AP2/ERF-ERF*, etc., with 278, 278, and 251 family members (Figure 5A). By analyzing the differential expression of all the TFs in five different tissue sections, it was found that 233 TFs were differentially expressed in the T vs. R group, 386 TFs were differentially expressed in the T vs. S group, 799 TFs were differentially expressed in the T vs. L group, and 508 TFs were differentially expressed in the T vs. F group. This investigation was based on the following criteria: |log2^FoldChange^| ≥ 1 and FPKM ≥ 50 (Figure 5B).

To screen modules related to rhizome development, a co-expression network was constructed using WGCNA, with a soft threshold of 0.8 (Appendix A). All the genes were divided into 41 modules of different colors (Figure 5C,D); among them, the red module had a significant correlation with rhizomes, with 1596 genes in the red module (Figure 5E,F). Then, after combining these with FPKM, a randomly selected group from the top 100 TFs differentially expressed in rhizomes was subjected to qRT-PCR. The expression of *BZIP1*, *C3H31*, *DREB12,* and *ERF114* was the most significant in rhizomes (T), *DREB21* was primarily expressed in the leaf (L), and *ERF41* was mainly expressed in the rhizomes (T) and roots (R) (Figure 5G).

To further elucidate the TFs may regulate the flavonoid biosynthesis pathways of *P. cyrtonema*, the correlation analysis of the flavonoid content in different tissues and the quantitative real-time PCR (qRT-PCR) results of TFs was analyzed (Figure 3B and Appendix A). The flavonoid content of rhizome was the richest in different tissues of *P. cyrtonema*. Meanwhile, the correlation analysis illustrated, the expression of *BZIP1*, *C3H31*, *DREB12*, and *ERF114* was correlated positively with the flavonoid content in different tissues, suggesting *BZIP1*, *C3H31*, *DREB12,* and *ERF114* may be involved in the flavonoid biosynthesis pathways of *P. cyrtonema*.

## 3. Discussion

The study of flavonoid metabolites has recently attracted extensive attention, especially in terms of *P. cyrtonema*, *P. kingianum,* and *P. sibiricum* [11,16,38]. The flavonoid in rhizomes of *P. cyrtonema* exhibits aging-delaying, blood-glucose-lowering, blood-lipid-lowering, atherosclerosis-preventive, and antimicrobial properties, and the flavonoid content directly impacts the quality of *P. cyrtonema*. Conducted to further investigate the flavonoid biosynthesis pathway in *P. cyrtonema*, an analysis of the transcriptomes from different tissues of the three-year-old *P. cyrtonema* plants showed continuous flavonoid accumulation continuously in rhizomes with the growth of *P. cyrtonema*. The types and relative content of flavonoid-biosynthesis-related genes were different from those in the leaves, roots, stems, and fruits. Most flavonoids, alkaloids, and amino acids were found to be the highest in the three-year age section rhizome (CT) compared with the one-year age section rhizome (AT) and the two-year age section rhizome (BT); this finding is consistent with that reported by Qingshuang Wang et al. [37].

The rhizome is a vital storage organ in *P. cyrtonema*, with high secondary metabolism. The secondary metabolism in rhizomes is significant for the growth and development of *P. cyrtonema*. The flavonoid content is rich in rhizome. The flavonoid content of three-year age section rhizome was richer significantly than one-year age section rhizome and two-year age section rhizome, and the flavonoid content of rhizome was the richest in different tissues of *P. cyrtonema*. The growth and development of rhizomes of *P. cyrtonema* are accompanied by accumulation of flavonoids. Meanwhile, the flavonoid content promoted the expansion and proliferation of rhizomes [37].

Meanwhile, 241 genes annotated to the flavonoid biosynthesis pathway were identified from different tissues of the three-year-old *P. cyrtonema*; among these genes, *PAL3*, *4CL3*, *CHI1*, *CHI1,* and *C4H* were significantly more expressed in the three-year age section (CT) than the one-year age section (AT) and the two-year age section (BT). The expression of these genes also affected the flavonoid content of the three-year age section (CT) directly. Using KEGG enrichment analysis, Anlin Li et al. [38] found that a differential metabolite in the rhizomes of *P. cyrtonema*, *P. kingianum*, and *P. sibiricum* showed that these genes were involved in flavonoid biosynthesis. Moreover, Junhong Chai et al. [15] found six DEGs involved in flavonoid biosynthesis, producing results identical to those reported in the present study.

The flavonoid biosynthesis pathways in *P. cyrtonema* are synergized by a series of structural and regulatory genes, including the structural genes in the flavonoid biosynthesis pathways, regulated by the transcription factors *MYB*, *bHLH*, *NAC*, and *WD40* [39]. In the three-year rhizomes of *P. cyrtonema*, differentially expressed TFs such as *MYB*-related, *WRKY*, *AP2/ERF-ERF*, *C3H*, *C2H2*, *bHLH*, *NAC*, *bZIP* and *MYB* may be involved in the development of the rhizomes of *P. cyrtonema*. Among them, the *MYB* family is the largest family in the *P. cyrtonema*. The number of *MYB*-related transcription factors was the highest in the rhizomes of *P. cyrtonema*, accounting for 5.24% of the total TFs. In addition, *MYB* family members have been found to regulate flavonoid biosynthesis pathways in most species, including *Arabidopsis thaliana* [40], *Fagopyrum tataricum* [29], *Petunia hybrida* [35], *Citrus* [32], *Fragum* [41], *Prunus avium L.* [25], and other species. Therefore, we speculate that *MYB* transcription factors are significant regulators of the flavonoid biosynthesis pathways of *P. cyrtonema*, constituting a significant role for promoting the development of the rhizomes of *P. cyrtonema*, improving resistance and secondary metabolites.

## 4. Materials and Methods

### 4.1. Plant Material

*P. cyrtonema* plants were collected from the *Polygonatum* breeding base in Shaowu City, Fujian Province; this base is part of the Sanming Academy of Agricultural Sciences, Fujian Province. Five different tissues, namely, fruit, leaf, root, stalk, and rhizome tissue, were collected from three-year-old *P. cyrtonema* plants. The materials for different age section rhizomes (AT: the one-year age section rhizome, BT: the two-year age section rhizome, and CT: the three-year age section rhizome) were obtained from *P. cyrtonema* plants and harvested in 9 October 2021. All the samples were collected for use in metabolite detection and RNA extraction, quickly frozen in liquid nitrogen, and stored at −80 °C in a refrigerator. Three plants were randomly selected as a mixed sample for each material, and the experiment was carried out in three biological replications.

### 4.2. Transcriptome Sequencing, Assembly, and Functional Annotation

The total RNA from five different tissues (fruit, leaf, root, stem, and rhizome) of three biological replicates was extracted using Trizol (Invitrogen, Carlsbad, CA, USA) kit. 15 cDNA libraries were constructed using Illumina’s NEBNext^®^UltraTMRNALibraryPrepKit, and, after the quality control was completed, data analysis of these 15 samples was carried out using the MyviCloud online platform. The Illumina transcriptome–sequencing data of these 15 samples were obtained using the IlluminaHiSeq platform produced by Tver Biotechnology Ltd., (Tver, Russia), and the data were analyzed using the MyviCloud online platform (https://cloud.metware.cn/, accessed on 30 November 2022). The raw data of these 15 samples were filtered to remove junctions using fastpv0.19.3. Q-value differential expression analysis of genes was conducted on 5 different tissues using DESeq2 software (NWXS–20-2243D-40), and the raw counts were statistically tested and corrected for hypothesis-testing probability (*p*-value) using the Benjamini and Hochberg method. The Bowtie method was used to clean bases, and the RESM method was used to calculate gene expression and FPKM values based on gene length.

### 4.3. Differential Expression Analysis of Genes

Differentially expressed genes in five different tissues were determined using DESeq2 software. Raw counts were tallied and corrected for hypothesis-testing probability (*p*-value) using the Benjamini and Hochberg method. |log2^FoldChange^| ≥ 1 and FDR < 0.05 were used as the screening conditions for DEGs.

### 4.4. Analysis of the Relative Flavonoid Content of Rhizomes of Different Age Sections

Differential metabolites were screened based on a combination of univariate statistics and multivariate statistical analyses. Based on the results of orthogonal partial least squares discriminant analysis (OPLS-DA), the variable importance projection (VIP) of the multivariate analysis OPLS-DA model obtained was used to screen the differential metabolites of different age sections based on the following criterion: VIP ≥ 1. Then, the difference multiplicity values (FC, FC ≥ 2, or FC ≤ 0.5) from the univariate analysis were combined to further screen the differential metabolites in the different age sections of *P. cyrtonema*. Meanwhile, the secondary classification of samples was used as the criteria for the identification of substances, and the different differentially flavonoid substances were screened to obtain the proportions of different age sections of *P. cyrtonema*, which were compared with the content of different age sections of *P. cyrtonema* using a difference heat map. The flavonoid content of the *P. cyrtonema* samples was assayed using a flavonoid content assay kit (Comin Biotechnology, Suzhou, China) according to the kit instructions: The absorbance at 510 nm was measured with a multifunctional enzyme marker (infinite M200 PRO, Tecan, Mannedorf, Switzerland).

### 4.5. Identification of Flavonoid-Biosynthesis-Related Genes in P. cyrtonema

In the transcriptome database of the different tissues (fruit, leaves, stems, roots, and rhizomes) of *P. cyrtonema*, the flavonoid-biosynthesis-related genes and amino acid sequences of Arabidopsis thaliana *PAL*, *C4H*, and *4CL* family genes were downloaded from the TAIR database (https://www.arabidopsis.org/, accessed on 1 November 2023). To identify the *PAL*, *C4H*, and *4CL* family members of *P. cyrtonema*, the amino acid sequences of the *PAL*, *C4H*, and *4CL* family genes of Arabidopsis thaliana were used as controls. With reference to the transcriptome data of different tissue parts of *P. cyrtonema*, TBtools v1.108 software (parameter settings: E-value—≤ 1 × 10^−5^; Number of Hits—500) was used to conduct local BLAST comparative searches to preliminarily screen for the family members of *P. cyrtonema PAL*, *C4H*, and *4CL*. Then, NCBI (https://www.ncbi.nlm.nih.gov/, accessed on 11 November 2023) and HMMER (https://www.ebi.ac.uk/Tools/hmmer/, accessed on 11 November 2023) were used to identify all the candidate sequences, and those with complete conserved domains were selected as family members of the *PAL*, *C4H*, and *4CL* genes of *P. cyrtonema*.

### 4.6. Validation of Transcript Levels of Differentially Expressed Genes

RNA was extracted from five tissues (fruit, leaf, root, stem, and rhizome) from the different age sections (AT, BT, and CT) using the RNAprep Pure Plant Plus Kit (Polysaccharides&Polyphenolics-rich) RNAprep Pure Kit (TIANGEN, Beijing, China), and cDNA first strands were synthesized via reverse transcription using a PerfectStart^®^ Uni RT&qPCR Kit (TransGen Biotech, Beijing, China). The cDNA was diluted 10-fold and used as a template, and the top100 TFs that were differentially expressed in the rhizomes were screened. Heatmaps (with a parameter of log_2_) were developed using TBtools v1.108 software to randomly select 6 transcription factor genes and flavonoid-biosynthesis-related genes differentially expressed in the rhizomes.

Primers were designed using the DNAMAN software (V6 6.0.3.99) produced by Prime Biology (https://primer3.ut.ee/, accessed on 14 November 2022). The expression of each gene was detected using qRT-PCR, and UBQ-E2-10 was utilized as the internal reference gene [42]. The reaction system amounted to 20 μL (SYBR Ⅱ (10 μL), ddH2O (6.4 μL), cDNA (2 μL), and (0.8 μL) each of upstream and downstream primers), and the relative expression of each gene was calculated according to 2^−ΔΔCt^. Three biological replicates were performed for each sample.

### 4.7. WGCNA Construction and Identification of Specific Modules

Weighted correlation network analysis (WGCNA) was performed using the WGCNA R Package (v1.69). The background correction and normalization of gene expression data was conducted to filter genes with low expression and small coefficients of variation by classifying the transcriptome data of different tissues in *P. cyrtonema*, constructing gene expression networks corresponding to different tissues of *P. cyrtonema* (Appendix A). The WGCNA was developed based on clustering relationships between genes. Co-expression modules related to rhizome development were screened by correlating different modules with different tissues.

### 4.8. Data Analysis

Significance analysis was performed using one-way ANOVA via IBM SPSS Statistics 26 software (SPSS, Inc., Chicago, IL, USA). Bar charts were plotted based on the data using GraphPad Prism 8 software (GraphPad, San Diego, CA, USA), and heatmaps were plotted using TBtools (TBtools v1.098769, Guangzhou, China). The correlation analysis of flavonoid content and the expression of TFs in different tissues of *P. cyrtonema* was performed with MWY (https://cloud.metware.cn/, accessed on 22 September 2023).

## 5. Conclusions

In summary, this study used tissues to generate high-resolution transcriptome datasets, including different tissues of fruits (F), leaves (L), roots (R), stems (S), and rhizomes (T) of three-year-old *P. cyrtonema* plants. To articulate the accumulation of the flavonoid biosynthesis pathways. The flavonoid content was increasing gradually from the one-year age section rhizome to the three-year age section rhizome, and the flavonoid content in different tissues was rich significantly in rhizome. The flavonoid-biosynthesis-related genes were mainly expressed in the three-year rhizomes, indicating differences in flavonoid accumulation in five different tissues and in different age section rhizomes. The results of quantitative real-time PCR (qRT-PCR) and the correlation analysis of flavonoid content and the expression of TFs in different tissues further indicated that *BZIP1*, *C3H31*, *ERF114*, and *DREB21* were differentially expressed in rhizomes, the DEGs may be involved in the accumulation of flavonoid in rhizome of *P. cyrtonema*.

## Figures and Tables

**Figure 1 plants-13-01524-f001:**
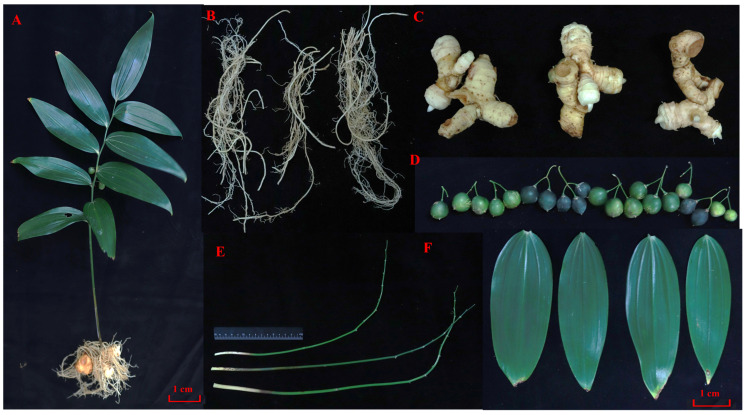
(**A**–**F**) Sampling of five tissue sites of *P. cyrtonema* ((**A**): whole plant, (**B**): root, (**C**): rhizome, (**D**): fruit, (**E**): stalk, and (**F**): leaf).

**Figure 2 plants-13-01524-f002:**
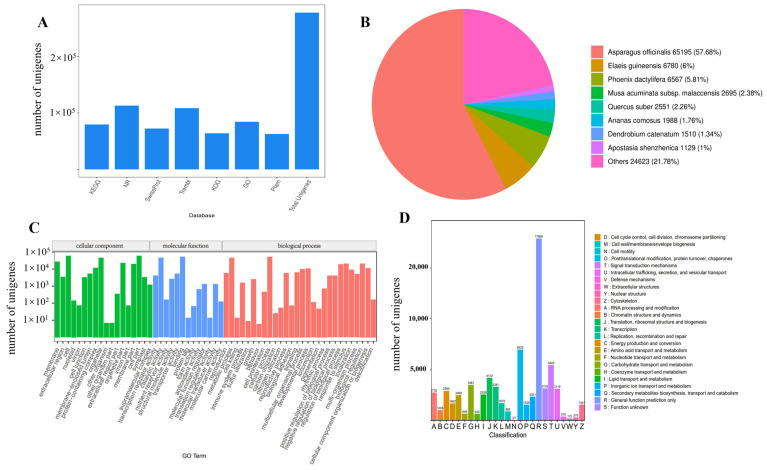
(**A**) Unigene annotation statistics; (**B**) NR annotations for species; (**C**) gene ontology (GO) function classification; (**D**) eukaryotic orthologous groups (KOG) function classification.

**Figure 3 plants-13-01524-f003:**
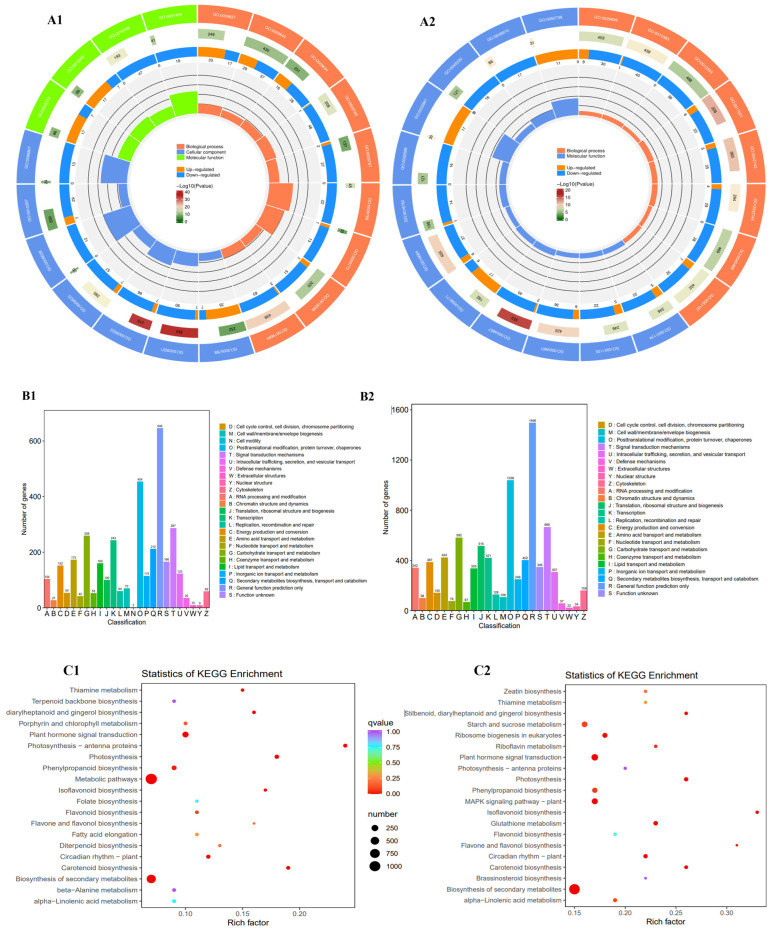
(**A1**) GO classification of differentially expressed genes (DEGs) in the F vs. T group; (**B1**) KOG classification of differentially expressed genes (DEGs) in the F vs. T group; (**C1**) top 20 KEGG pathways among the DEGs in the F vs. T group. (**A2**) GO classification of differentially expressed genes (DEGs) in the L vs. T group; (**B2**) KOG classification of differentially expressed genes (DEGs) in the L vs. T group; (**C2**) top 20 KEGG pathways among the DEGs in the L vs. T group. (**A3**) GO classification of differentially expressed genes (DEGs) in the R vs. T group; (**B3**) KOG classification of differentially expressed genes (DEGs) in the R vs. T group; (**C3**) top 20 KEGG pathways among the DEGs in the R vs. T group. (**A4**) GO classification of differentially expressed genes (DEGs) in the S vs. T group; (**B4**) KOG classification of differentially expressed genes (DEGs) in the S vs. T group; (**C4**) top 20 KEGG pathways among the DEGs in the S vs. T group. (**D**) K-means clustering analysis of all DEGs. The fragments per kilobase of transcript per million mapped reads (FPKMs) of all the DEGs were used for K-means analysis. (**E**) A clustered heat map of all the samples. (**F**) Venn diagram of the four groups, F vs. T, L vs. T, R vs. T, and S vs. T (F: fruit, L: leaf, R: root, S: stalk, and T: rhizome).

**Figure 4 plants-13-01524-f004:**
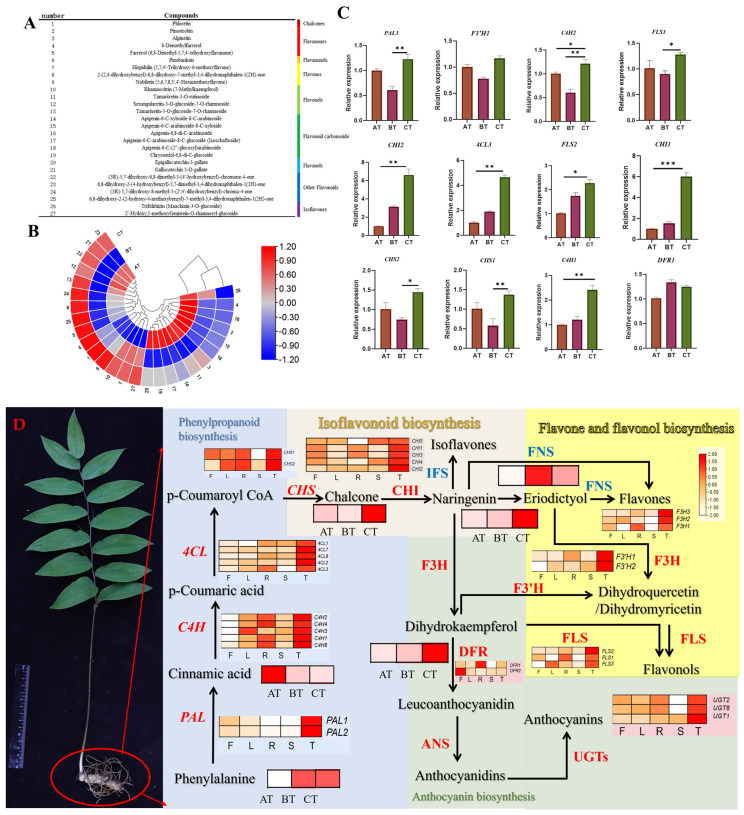
(**A**) 27 Flavonoids in the rhizomes of *P. cyrtonema*; (**B**) 27 flavonoid proportions of different age section rhizomes in *P. cyrtonema*; (**C**) the expression of flavonoid-biosynthesis-related genes in different age section rhizomes of *P. cyrtonema*; (**D**) variation in flavonoid accumulation in different tissues of *P. cyrtonema* (AT: the one-year age section rhizome, BT: the two-year age section rhizome, and CT: the three-year age section rhizome), (F: fruit, L: leaf, R: root, S: stalk, and T: rhizome), (Student’s *t*-test, * *p* < 0.05, ** *p* < 0.01, *** *p* < 0.001), (The arrows represent regulatory mechanisms).

**Figure 5 plants-13-01524-f005:**
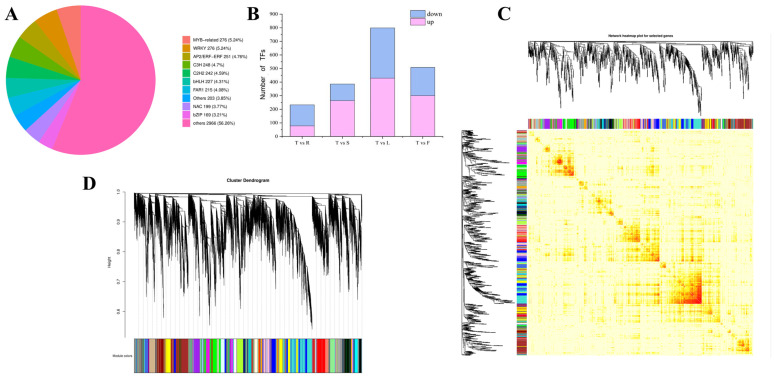
(**A**) All transcription factors (TFs) in *P. cyrtonema.* (**B**) Transcription factor expression patterns in the F vs. T, L vs. T, R vs. T, and S vs. T groups. (**C**) Transcriptome analysis based on WGCNA, displaying the connectivity between genes. (**D**) Clustering analysis based on WGCNA, with 41 modules of different colors. (**E**) Correlation analysis between modules and different tissues, namely, fruits (F), leaves (L), roots (R), stems (S), and rhizomes (T), in the WGCNA. (**F**) Heatmap of the red module in the WGCNA analysis of different tissues. (**G**) Expression analysis of the genes *BZIP1*, *C3H31*, *ERF114*, and *ERF41* in different tissues (F: fruit, L: leaf, R: root, S: stalk, and T: rhizome), (Student’s *t*-test, ** *p* < 0.01, *** *p* < 0.001).

## Data Availability

Data are contained within this article and its Appendix A. Raw (unprocessed) data are available on request from the corresponding author.

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
