# Peer review of "Transcriptomic Analysis Reveals the Flavonoid Biosynthesis Pathway Involved in Rhizome Development in Polygonatum cyrtonema Hua"

_plants, 2024, doi:10.3390/plants13111524_

Round 1
Reviewer 1 Report
Comments and Suggestions for Authors
Kui Wan et al has reported the manuscript “Transcriptomic Analysis Reveals the Flavonoid Biosynthesis Pathway Involved in Rhizome Development in Polygonatum cyrtonema Hua”. In this manuscript, researchers have used bioinformatic tools and gene expression validation by qRT-PCR to study flavonoid biosynthesis-related genes in different tissues of Polygonatum cyrtonema Hua.
I see some aspects which should be improved before the manuscript is accepted for publication.
Major comments:
Q1. The manuscript should be edited by English editing institution
The manuscript needs strict language editing.
Minor comments:
Q2. Line 81, “characterized in red module”. What is meaning of “red Module”
Q3. Line 80, “AP2/ERF et al.”, How “et al.”. It looks something is missing here.
There are so many such discrepancies in the manuscript. Please rewrite the manuscript removing errors.
Comments on the Quality of English LanguageThe manuscript needs language editing.
Author Response
Dear reviewer:
Thank you very much for reviewing our manuscript and for your professional comments. We appreciate your time and effort in helping us to improve the quality and readability of our manuscript. We have corrected and improved the manuscript according to your comments:
Q1. The manuscript should be edited by English editing institution. The manuscript needs strict language editing.
Thank you for pointing out this problem. We've had a professional make the correction.
Q2. Line 81, “characterized in red module”. What is meaning of “red Module”
About the red Module, we explained it in the line 252 to line 257 of the manuscript
“To screen modules related to rhizome development, a co-expression network was constructed using WGCNA, with a soft threshold of 0.8 (Figure S3 A). All the genes were divided into 41 modules of different colors (Figure 5 C and D); among them, the red module had a significant correlation with rhizomes, with 1596 genes in the red module (Figure 5 E and F ), and, combined with FPKM, a randomly selected group from the top 100 TFs differentially expressed in rhizomes were subjected to qRT-PCR.”
Q3. Line 80, “AP2/ERF et al.”, How “et al.”. It looks something is missing here. There are so many such discrepancies in the manuscript. Please rewrite the manuscript removing errors.
Thank you for your question. We've had a professional make the correction.
line 86-88 “Nine differentially expressed transcription factors family (MYB, WRKY, AP2/ERF, C3H, bHLH, NAC, bZIP, C2H2 and FAR1) were characterized in the red module.
Thank you very much for your attention and time. Look forward to hearing from you.
Yours sincerely
Reviewer 2 Report
Comments and Suggestions for Authors
This study seems interesting, but the writing makes it hard to follow the story. I strongly recommend the authors improve their writing significantly before resubmitting.
For example, lines 13-16 must be reorganized to make the sentence more informative.
line 19, "Combined" was used.
line 21, more information is required for "in red module".
line 30, "and" was used incorrectly.
line 34-36, should be split to two sentences.
line 63-64, not readable.
line 74-84, it is difficult to understand such a so long sentence.
Comments on the Quality of English Language
Need to be improved significantly.
Author Response
Dear reviewer:
Thank you very much for reviewing our manuscript and for your professional comments. We appreciate your time and effort in helping us to improve the quality and readability of our manuscript. We have corrected and improved the manuscript according to your comments:
- lines 13-16 must be reorganized to make the sentence more informative.
Thank you for your comments. We've made correction in the manuscript.
lines 13-17 “To articulate the molecular mechanisms the of flavonoid biosynthesis pathway, we examined transcriptome changes using Illumina HiSeq from five different tissues and the RNA-seq of 15 samples generated over 105 Gb of a, generating a total of 277,955 unigenes. The cDNA libraries of the fruits (F), leaves (L), roots (R), stems (S), and rhizomes (T) of three-year-old P. cyrtonema plants generated 57,591, 53,578, 60,321, 51,530, and 54,935 unigenes.”
- line 19, "Combined" was used.
We agree with this comment. Therefore, we have modified this point.
line 20 “In addition, combined with WGCNA and the FPKM of five tissues transcription, nine differentially expressed transcription factor families (MYB, WRKY, AP2/ERF, etc.) were characterized in the red module.”
- line 21, more information is required for "in red module".
About the red Module, we explained it in the line 252 to line 257 of the manuscript
“To screen modules related to rhizome development, a co-expression network was constructed using WGCNA, with a soft threshold of 0.8 (Figure S3 A). All the genes were divided into 41 modules of different colors (Figure 5 C and D); among them, the red module had a significant correlation with rhizomes, with 1596 genes in the red module (Figure 5 E and F ), and, combined with FPKM, a randomly selected group from the top 100 TFs differentially expressed in rhizomes were subjected to qRT-PCR.”
- line 30, "and" was used incorrectly.
Thank you for your comments. We've made correction in the manuscript.
line 32 “Polygonatum cyrtonema Hua (P. cyrtonema) is an important Chinese herb with homology between medicine and food with respect to Polygonatum Mill (Liliaceae), which isincluded in the Chinese Pharmacopoeia as a basal species of the Chinese medicine Polygonatum, together with P.sibiricum and P.kingianum.”
- line 34-36, should be split to two sentences.
We agree with this comment. Therefore, we have modified this point.
line 36-39 “Flavonoids and polysaccharides are the main active ingredients of P. cyrtonema rhizomes [3]. Flavonoids and polysaccharides are important indicators of the herb quality of P. cyrtonema rhizomes [4].”
- line 63-64, not readable.
Thank you for your comments. We've made correction in the manuscript.
line 68-69 “Therefore, it is crucial to study the regulation mechanism of the flavonoid biosynthesis pathway of Polygonatum.”
- line 74-84, it is difficult to understand such a so long sentence.
Thank you for your question. We've had a professional make the correction.
line 80-91 “At present, the lack of genomic information hinders the systematic study of the flavonoid-biosynthesis-related genes of P. cyrtonema. To elucidate the molecular mechanisms of flavonoid biosynthesis pathways in the rhizomes of P. cyrtonema, we conducted a transcriptomic analysis of different tissues, including, fruits, leaves, roots, stems, and rhizomes, of three-year-old P. cyrtonema plants using Illumina RNA-seq technology. Meanwhile, combined with WGCNA and the FPKM of five tissues’ transcription, nine differentially expressed transcription factor families (MYB, WRKY, AP2/ERF, C3H, bHLH, NAC, bZIP, C2H2, and FAR1) were characterized in the red module. Quantitative real-time PCR (qRT-PCR) further indicated that BZIP1, C3H31, ERF114, DREB21, and DREB12 are involved in rhizome development in P. cyrtonema, providing insight into the molecular and developmental mechanisms of the rhizomes of P. cyrtonema.”
Thank you very much for your attention and time. Look forward to hearing from you.
Yours sincerely
Reviewer 3 Report
Comments and Suggestions for Authors
In the paper “Transcriptomic Analysis Reveals the Flavonoid Biosynthesis Pathway Involved in Rhizome Development in Polygonatum cyrtonema Hua”, the authors show a research design for the study of metabolic pathways related to rhizome development in Polygonatum Cyrtonema Hua.
COMMENTS:
The authors’ results support the identification of genes involved in the rhizome development of P. Cyrtonema. However, the introduction and discussion section lack information or discussion about the significance of these results. Instead, they primarily focus on flavonoid biosynthesis in general or the potential pharmacological effect of the flavonoids, which diminishes the importance of the paper. Therefore, it is recommended that the authors enhance the significance of their manuscript by providing a comprehensive introduction section and more focused discussion that supports their results related to rhizomes development. Among other potential applications, this data could contribute to increasing flavonoid production by improving developmental characteristics of rhizomes, such as size or number, in P. Cyrtonema.
Overall, the writing style is confusing. Here some suggestions:
The authors conducted analysis at different ages (AT, BT, and CT), but, they do not clearly specify that these sections were done from rhizome. It is recommended that the authors specify the location, including the word “rhizome” in each phrase.
The extensive list of functional category of genes was described in figure 3 and S2. Therefore, it is recommended to summarize the list of genes in paragraph 134-186.
There are confusing phrases throughout the manuscript thar require that the authors’ attention. (e.g 217-224).
The authors mentions that Raw (unprocessed) data are available on request from the corresponding author. However, given that published data should be easily verifiable by any research group, it is customary to deposit databases in a public repository, such as NCBI.
The manuscript suggests that the found results were not the main objective at the beginning of the project, which decreases the significance of the results. Additionally, the disorganized paragraph and poor results description contribute to a confusing manuscript. However, the authors present interesting findings.
Author Response
Dear reviewer:
Thank you very much for reviewing our manuscript and for your professional comments. We appreciate your time and effort in helping us to improve the quality and readability of our manuscript. We have corrected and improved the manuscript according to your comments:
- Therefore, it is recommended that the authors enhance the significance of their manuscript by providing a comprehensive introduction section and more focused discussion that supports their results related to rhizomes development. Among other potential applications, this data could contribute to increasing flavonoid production by improving developmental characteristics of rhizomes, such as size or number, in P. Cyrtonema.
Thank you for your comments. We've made correction in the manuscript.
Lines 35-36: “The rhizome is a vital storage organ in P. cyrtonema, with high metabolic process.”
Lines 79-81: “the flavonoid content in different age section rhizomes and different tissues, indicated the accumulation of flavonoid may be involved in rhizome development in P. cyrtonema.”
Lines 293-300: “The rhizome is a vital storage organ in P. cyrtonema, with high secondary metabolism. The secondary metabolism in rhizomes is significant for the growth and development of P. cyrtonema. The flavonoid content is rich in rhizome. The flavonoid content of three-year age section rhizome was richer significantly than one-year age section rhizome and two-year age section rhizome, and the flavonoid content of rhizome was the richest in different tissues of P.cyrtonema. The growth and development of rhizomes of P. cyrtonema are accompanied by accumulation of flavonoids. Meanwhile, the flavonoid content promoted the expansion and proliferation of rhizomes [37].”
- The authors conducted analysis at different ages (AT, BT, and CT), but, they do not clearly specify that these sections were done from rhizome. It is recommended that the authors specify the location, including the word “rhizome” in each phrase.
We agree with this comment. Therefore, we have specified the location in each phrase.
Lines 203-205: “Based on the broadly targeted metabolomes of different age section rhizomes (AT: the one-year age section rhizome, BT: the two-year age section rhizome, and CT: the three-year age section rhizome) obtained in the previous stage [37].”
Lines 424-428: “The flavonoid content was increasing gradually from the one-year age section rhizome to the three-year age section rhizome, and the flavonoid content in different tissues was rich significantly in rhizome. The flavonoid-biosynthesis-related genes were mainly expressed in the three-year rhizomes, indicating differences in flavonoid accumulation in five different tissues and in different age section rhizomes.”
- The extensive list of functional category of genes was described in figure 3 and S2. Therefore, it is recommended to summarize the list of genes in paragraph 134-186.
Thank you for your comments. We have summarized this point.
Lines 172-186: “In total, for the four groups, GO enrichment mostly corresponded to “fructosyltransferase activity”, “photosynthesis”, and “enzyme inhibitor activity”. KOG enrichment for the four groups was mainly in relation to R (for general function prediction only), O (for post-translational modification, protein flipping, and chaperones) and T (for signal transduction mechanisms). The four groups’ KEGG enrichment was different in relation to “Metabolic pathways”, “Flavonoid biosynthesis”, “Anthocyanin biosynthesis”, and “Isoflavonoid biosynthesis”.
Meanwhile, K-means clustering analysis showed that the clustering of genes with high expression was different in the five samples (Figure 3 D and E). A total of 22,263 DEGs were identified in the F vs T, L vs T, R vs T, and S vs T groups; among them, 379 DEGs were in the four groups, and 181 DEGs were highly expressed in rhizomes (Figure 3 F), mainly corresponding to “Metabolic pathways” and “Flavonoid biosynthesis” in the major databases. Additionally, it was speculated that these DEGs may be the key factors for revealing the flavonoid biosynthesis pathways involved in rhizome development in P. cyrtonema.”
- There are confusing phrases throughout the manuscript thar require that the authors’ attention. (e.g 217-224).
Thank you for your comments. We've made correction in the manuscript.
Lines 172-186: “Meanwhile, the flavonoid content was increasing gradually from the one-year age section rhizome to the three-year age section rhizome (Figure S3A). the flavonoid content in different tissues was rich significantly in rhizome (Figure S3B). The changes provided data supporting the flavonoid biosynthesis pathways may be involved in rhizome development in P. cyrtonema.”
- The authors mentions that Raw (unprocessed) data are available on request from the corresponding author. However, given that published data should be easily verifiable by any research group, it is customary to deposit databases in a public repository, such as NCBI.
- Thank you for your comments, we agree with this comment. We're collating the transcriptome data, and after that, we'll upload the data.
Thank you very much for your attention and time. Look forward to hearing from you.
Yours sincerely

Reviewer 4 Report
Comments and Suggestions for Authors
This is an interesting investigation described gene accompanied flavonoid photosynthesis in mature rhizomes of different ages. Authors present huge amount of transriptomes materials, and it primary analysis. Transcriptions factor and DEG comparison was done for different organs and different ages. Actually, it is very logical that rhizome as “storage organs”, not primary and not transport one) have a high secondary metabolism. It will be great to discused at least in few words this point.
Despite interesting topic, the current text require significant modification. First of all, to claim “molecular mechanism” authors need to generate mutants for each gene and study the effect of these gene on rhizome development: see title: “flavonoid biosynthesis pathway invloved in rhizome development” can not be concluded from simple trascriptomic analysis.
Title need modifications.
Line 9: rhizomes rich in flavonoids, not abundant!
Many words were fused in Abstract and in whole text. Please, add space between.
Line 24: are involved = are accompanied.
Ine 37: primary active ingredients ? What is primary active?
Lines 47 – 49: split sentence to two.
Lines 79 – 84: copy from abstracts.
Line 134: “6,838 differential genes” - may be differently expressed genes?
Line 283: Plants can not exhibit properties itself. Extract form some organs etc can
Line 295: more abundant significantly” ?? Please, check similar statements through whole text and edit.
Line 308: “transcriptomics studies in flavonoids???
Lines 319- 320: for calim invlovemnet many other studies are required like up of down regulation of each gene(and combination) and it effcet on transcriptome amd rhizome developments.
Line 336: what is age section? Why section?
Line 357: please, edit.
Line 373: secondary mass spectrometry”? What is the primary one?
Comments on the Quality of English Language
Require significant corrections: punctuations, fused words, sentences structure, wrong terms.
Author Response
Dear reviewer:
Thank you very much for reviewing our manuscript and for your professional comments. We appreciate your time and effort in helping us to improve the quality and readability of our manuscript. We have corrected and improved the manuscript according to your comments:
- Title need modifications. to claim “molecular mechanism” authors need to generate mutants for each gene and study the effect of these gene on rhizome development: see title: “flavonoid biosynthesis pathway invloved in rhizome development” can not be concluded from simple trascriptomic analysis.
- Thank you for your comments. we have modified this “molecular mechanism” in the manuscript.
Lines 9-12: “However, a comprehensive overview of Polygonatum flavonoid biosynthesis pathways is still absent. To articulate the accumulation of the flavonoid biosynthesis pathways, we examined transcriptome changes using Illumina HiSeq from five different tissues and the RNA-seq of 15 samples had over 105 Gb of a clean base, generating a total of 277,955 unigenes.”
Lines 24-25: “Therefore, this study provides a foundation for further research into uncovering the accumulation of flavonoid biosynthesis in the rhizomes of P. cyrtonema.”
Lines 429-434: “The results of quantitative real-time PCR (qRT-PCR) and the correlation analysis of flavonoid content and the expression of TFs in different tissues further indicated that BZIP1, C3H31, ERF114, and DREB21 may be involved in rhizome development in P. cyrtonema, providing a research foundation for the further investigation of the accumulation of flavonoid biosynthesis pathways in the rhizomes of Polygonatum.”
- Line 9: rhizomes rich in flavonoids, not abundant!
Thank you for your comments. We've made correction in the manuscript.
Lines 9-12: “Polygonatum cyrtonema Hua (P. cyrtonema) rhizomes are rich in flavonoids and other secondary metabolites, exhibiting remarkable antioxidant, anti-tumor, and immunomodulatory effects. Polygonatum flavonoid-biosynthesis-related genes have been characterized by some researchers already.”
Lines 424-426: “The flavonoid content was increasing gradually from the one-year age section rhizome to the three-year age section rhizome, and the flavonoid content in different tissues was rich significantly in rhizome.”
- Many words were fused in Abstract and in whole text. Please, add space between.
We agree with this comment. Therefore, we have modified this point.
Lines 13-15: “To articulate the accumulation of the flavonoid biosynthesis pathways, we examined transcriptome changes using Illumina HiSeq from five different tissues and the RNA-seq of 15 samples had over 105 Gb of a clean base, generating a total of 277,955 unigenes.”
- Line 24: are involved = are accompanied.
Thank you for your comments. We have modified this point.
Line 22-24: “Quantitative real-time PCR (qRT-PCR) further indicated that BZIP1, C3H31, ERF114, and DREB21 are accompanied in rhizome development in P. cyrtonema.”
- Iine 37: primary active ingredients ? What is primary active?
Thank you for your comments. We've made correction in the manuscript.
Lines 35-37: “The rhizome is a vital storage organ in P. cyrtonema, with high metabolic process. Flavonoids and polysaccharides are the main medicinal ingredients of P. cyrtonema rhizomes.”
- Lines 47 – 49: split sentence to two.
We agree with this comment. Therefore, we have modified this point.
Lines 47–49: “Ye [11] used three- and nine-month rhizomes of P.cyrtonema to perform transcriptomic analysis. Consequently, 12 key genes involved in flavonoid biosynthesis pathways were identified.”
- Lines 79 – 84: copy from abstracts.
Thank you for your comments. We have modified this point.
Lines 79–84: “To elucidate the accumulation of flavonoid biosynthesis pathways in the rhizomes of P. cyrtonema, we conducted a transcriptomic analysis of different tissues, including, fruits, leaves, roots, stems, and rhizomes, taken from three-year-old P. cyrtonema plants using Illumina RNA-seq technology and the flavonoid content in different age section rhizomes and different tissues, indicated the accumulation of flavonoid may be involved in rhizome development in P. cyrtonema. Meanwhile, combined with WGCNA and the FPKM of five tissues’ transcription, quantitative real-time PCR (qRT-PCR) further indicated that BZIP1, C3H31, ERF114, DREB21, and DREB12 may be involved in rhizome development in P. cyrtonema.”
- Line 134: “6,838 differential genes” - may be differently expressed genes?
Thank you for your comments. We've made correction in the manuscript.
Lines 131–135: “GO, KOG, and KEGG enrichment analyses were conducted on four groups (F vs T, L vs T, R vs T, and S vs T) of DEGs (differentially expressed genes). There were 6,838 DEGs in F vs T; of these, 3,219 DEGs were upregulated, and 3,619 DEGs were downregulated.”
- Line 283: Plants can not exhibit properties itself. Extract form some organs etc can
We agree with this comment. Therefore, we have modified this point in the manuscript.
Lines 281– 284: “The flavonoid in rhizomes of P.cyrtonema exhibits aging-delaying, blood-glucose-lowering, blood-lipid-lowering, atherosclerosis-preventive, and antimicrobial properties, and the flavonoid content directly impacts the quality of P. cyrtonema.”
- Line 295: more abundant significantly” ?? Please, check similar statements through whole text and edit.
Thank you for your comments. we have modified this point in the manuscript.
Lines 293– 300: “The rhizome is a vital storage organ in P. cyrtonema, with high secondary metabolism. The secondary metabolism in rhizomes is significant for the growth and development of P. cyrtonema. The flavonoid content is rich in rhizome. The flavonoid content of three-year age section rhizome was richer significantly than one-year age section rhizome and two-year age section rhizome, and the flavonoid content of rhizome was the richest in different tissues of P.cyrtonema. The growth and development of rhizomes of P. cyrtonema are accompanied by accumulation of flavonoids. Meanwhile, the flavonoid content promoted the expansion and proliferation of rhizome [37].”
- Line 308: “transcriptomics studies in flavonoids???
We agree with this comment. we have deleted the point in the manuscript.
- Lines 319- 320: for calim invlovemnet many other studies are required like up of down regulation of each gene (and combination) and it effcet on transcriptome amd rhizome developments.
Thank you for your comments.
Combined with qRT-PCR and the FPKM of five tissues’ transcription, we speculated the differentially expressed TFs may be involved in the development of the rhizomes of P.cyrtonema.
Lines 314- 316: “In the three-year rhizomes of P.cyrtonema, differentially expressed TFs such as MYB-related, WRKY, AP2/ERF-ERF, C3H, C2H2, bHLH, NAC, bZIP and MYB may be involved in the development of the rhizomes of P.cyrtonema.”
- Line 336: what is age section? Why section?
- Thank you for your comments. we have modified this point in the manuscript.
Lines 331- 334: “The materials for different age section rhizomes (AT: the one-year age section rhizome, BT: the two-year age section rhizome, and CT: the three-year age section rhizome) were obtained from P. cyrtonema plants and harvested in October 2021.”
- Line 357: please, edit.
We agree with this comment. Therefore, we have modified this point in the manuscript.
Lines 350- 352: “The Bowtie method was used to clean bases, and the RESM method was used to calculate gene expression and FPKM values based on gene length.”
- Line 373: secondary mass spectrometry”? What is the primary one?
Thank you for your comments. We have modified this point in the manuscript.
Lines 366- 369: “Meanwhile, the secondary classification of the samples was used as the criteria for the identification of substances, and the different differentially flavonoid substances were screened to obtain the proportions of different age sections of P. cyrtonema.”
Thank you very much for your attention and time. Look forward to hearing from you.
Yours sincerely

Round 2
Reviewer 2 Report
Comments and Suggestions for Authors
As stated in the study, the authors applied transcriptomic analysis in order to understand the molecular mechanisms of flavonoid biosynthesis pathways in the rhizomes of P. cyrtonema. Several hundreds of genes were identified as differentially expressed through comparative analysis in different tissues using RNA-seq. The authors also utilized qRT-PCR but indicated only some of the identified genes from RNA-seq were upgraded while others were downregulated. Overall, the novelty of this work is limited with only gene identification work presented. To improve this study to fit the high quality of Journal Plants, the authors should think about how to validate the pathways/genes involved in flavonoid biosynthesis pathways.
Comments on the Quality of English Languageneed to be improved
Author Response
Dear reviewer:
Thank you very much for reviewing our manuscript and for your professional comments. We appreciate your time and effort in helping us to improve the quality and readability of our manuscript. We have corrected and improved the manuscript according to your comments:
- the authors should think about how to validate the pathways/genes involved in flavonoid biosynthesis pathways.
Thank you for your comments. We've made correction in the manuscript.
We added the flavonoid content of different age section (AT:the one-year age section, BT: the two-year age section, and CT: the three-year age section), and different tissues (F: fruit, L: leaf, R: root, S: stalk, and T: rhizome). The flavonoid content was abundant significantly in three-year-old rhizomes. Meanwhile, these up-regulated genes will also be the key genes for our future research.
lines 220-223 “and the flavonoid content was increasing gradually from the one-year age section to the three-year age section (Figure S3A). Meanwhile, the flavonoid content was abundant significantly in rhizome (Figure S3B). Among these, the up-regulated genes were involved in the flavonoid biosynthesis pathway of P. cyrtonema.”
lines 232-233 “the expression of these genes also affected the flavonoid content of rhizome in P. cyrtonema.”
lines 285-287 “The flavonoid content of three-year age section was more abundant significantly than one-year age section and two-year age section, and the flavonoid content of rhizome was most abundant in different tissues of P.cyrtonema.”
lines 325-329 “Five different tissues, namely, fruit, leaf, root, stalk, and rhizome tissue, were collected from three-year-old P. cyrtonema plants. The materials for all three age sections (AT, BT, and CT) of rhizomes were obtained from P. cyrtonema plants and harvested in October 2021. All the samples were collected for metabolite detection and RNA extraction.”
lines 325-329 “The flavonoid content of the P.cyrtonema samples was assayed using a flavonoid content assay kit (Comin Biotechnology, Suzhou, China) according to the kit instructions: The absorbance at 510 nm was measured with a multifunctional enzyme marker (infinite M200 PRO, Tecan, Switzerland Confederation).”
lines 436-437 “Figure S3: (A) the flavonoid content in different age sections of P. cyrtonema.
(B) the flavonoid content in different tissues of P. cyrtonema.”
Thank you very much for your attention and time. Look forward to hearing from you.
Yours sincerely

Reviewer 4 Report
Comments and Suggestions for Authors
The text is fine now.
Comments on the Quality of English LanguageMinor editing
Round 3
Reviewer 2 Report
Comments and Suggestions for Authors
The comments have not been solved properly. The key point here is how can you validate that the enhanced genes are the main regulator for flavonoid increase/synthesis. Why do authors use both next-generation sequencing and RT-PCR? do you plan to use RT-PCR to validate and confirm the identified genes?
Comments on the Quality of English LanguageN/A
Author Response
Dear reviewer:
Thank you very much for reviewing our manuscript and for your professional comments. We appreciate your time and effort in helping us to improve the quality and readability of our manuscript. We have corrected and improved the manuscript according to your comments:
- The comments have not been solved properly. The key point here is how can you validate that the enhanced genes are the main regulator for flavonoid increase/synthesis. Why do authors use both next-generation sequencing and RT-PCR? do you plan to use RT-PCR to validate and confirm the identified genes?
Thank you for your comments. We've corrected in the manuscript.
In the previous manuscript, we added the flavonoid content of different age section (AT:the one-year age section, BT: the two-year age section, and CT: the three-year age section), and different tissues (F: fruit, L: leaf, R: root, S: stalk, and T: rhizome). The flavonoid content was abundant significantly in three-year-old rhizomes. Meanwhile, we already had the heatmap expression and the quantitative real-time PCR (qRT-PCR) results of ERF41, DREB21, BZIP1, C3H31, DREB12, and ERF114 in different tissues of P. cyrtonema (Figure 5 G), these TFs were up-regulated differentially in rhizomes of P. cyrtonema. Therefore, in this manuscript, we added the correlation analysis of the flavonoid content and the quantitative real-time PCR (qRT-PCR) results of ERF41, DREB21, BZIP1, C3H31, DREB12, and ERF114 in different tissues of P. cyrtonema, suggesting BZIP1, C3H31, DREB12, and ERF114 may be involved in the flavonoid biosynthesis pathways of P. cyrtonema, and these up-regulated TFs will also be the key TFs for our future functional research. The corrections as follows:
Lines 256-260: The heatmap expression and the quantitative real-time PCR (qRT-PCR) results of ERF41, DREB21, BZIP1, C3H31, DREB12, and ERF114 in different tissues of P. cyrtonema (Figure 5 G).
Lines 262-269: “To further elucidate the TFs may regulate the flavonoid biosynthesis pathways of P. cyrtonema, the correlation analysis of the flavonoid content in different tissues and the quantitative real-time PCR (qRT-PCR) results of TFs was analyzed (Figure 3 B and Table S5). The flavonoid content of rhizome was most abundant in different tissues of P.cyrtonema. Meanwhile, the correlation analysis illustrated, the expression of BZIP1, C3H31, DREB12, and ERF114 was correlated positively with the flavonoid content in different tissues, suggesting BZIP1, C3H31, DREB12, and ERF114 may be involved in the flavonoid biosynthesis pathways of P. cyrtonema.”
Lines 429-431: “The correlation analysis of flavonoid content and the expression of TFs in different tissues of P.cyrtonema was performed with MWY(https://cloud.metware.cn/).”
Lines 434-437: “Nine TF families were found to be specifically expressed in rhizomes, and the results of quantitative real-time PCR (qRT-PCR) and the correlation analysis of flavonoid content and the expression of TFs in different tissues further indicated that BZIP1, C3H31, ERF114, and DREB21were involved in rhizome development in P. cyrtonema.”
Lines 457-458 “Table S5: The correlation analysis of the flavonoid content and the quantitative real-time PCR (qRT-PCR) results of TFs in different tissues of P. cyrtonema.
Lines 436-437 “Figure S3: (A) the flavonoid content in different age sections of P. cyrtonema.
(B) the flavonoid content in different tissues of P. cyrtonema.”
Thank you very much for your attention and time. Look forward to hearing from you.
Yours sincerely
